# Adverse Events in Open Surgical vs. Ultrasound-Guided Percutaneous Brachial Access for Endovascular Interventions

**DOI:** 10.3390/jcm13144179

**Published:** 2024-07-17

**Authors:** Evren Ozcinar, Nur Dikmen, Ahmet Kayan, Cagdas Baran, Levent Yazicioglu

**Affiliations:** 1Department of Cardiovascular Surgery, Heart Center, Cebeci Hospitals, Ankara University School of Medicine, Ankara 06230, Turkey; evrenozcinar@gmail.com (E.O.); cagdasbaran@gmail.com (C.B.); leventyazicioglu@gmail.com (L.Y.); 2Department of Cardiovascular Surgery, Kirikkale High Specialization Hospital, Kirikkale 71300, Turkey; dr.ahmet.kayan@gmail.com

**Keywords:** access site complications, ultrasound guidance, peripheral arterial disease, open surgical incision, endovascular interventions

## Abstract

**Background:** Advances in endovascular interventions have made endovascular approaches the first option for treating peripheral arterial diseases. Although radial artery access is commonly used for coronary procedures, the common femoral artery remains the most frequent site for endovascular treatments due to better ergonomics and proven technical success. Meanwhile, data on using upper extremity access via the brachial artery during complex endovascular aortic interventions are lacking. This study aimed to compare the incidence of access site complications between ultrasound-guided percutaneous brachial access (UPA) and open surgical incisional brachial access (OSA) in the management of peripheral arterial diseases. **Methods:** Patients who underwent treatment for peripheral arterial and aortic disease using brachial access from 2019 to 2023 were included in this study. The primary endpoint was the complication rate at the access site 30 days postoperatively. Access-related complications included bleeding requiring re-exploration, acute upper limb ischemia, thrombosis, pseudoaneurysm, arteriovenous fistula, and nerve injury associated with the brachial access. **Results:** Brachial access was performed on 485 patients (UPA, *n* = 320; OSA, *n* = 165). The mean operation time was 164.5 ± 45.4 min for the percutaneous procedure and 289.2 ± 79.4 min for the cutdown procedure (*p* = 0.003). Postprocedural hematoma occurred in 15 patients in the UPA group and 2 patients in the OSA group (*p* = 0.004). Thromboembolic events were observed in 9 patients in the percutaneous group and 3 patients in the OSA group. Reoperation was required for 23 patients in the percutaneous group and 8 patients in the cutdown group. **Conclusions:** The findings indicate that patients undergoing endovascular arterial interventions have a higher rate of brachial access complications in the UPA group compared to the OSA group.

## 1. Introduction

Advances in endovascular interventions have made endovascular approaches the first choice for treating peripheral arterial diseases [1]. Although radial artery access is preferred for coronary artery procedures, the common femoral artery remains the most commonly used site for endovascular treatments in peripheral arterial interventions [2]. This preference is due to the superior ergonomics and proven technical success associated with femoral access, leading to a significant portion of peripheral and aortic pathologies being managed through this route. However, brachial access is emerging as a viable alternative, especially for lesions involving renovisceral arteries, given the downward and angulated pattern of this arterial tree [3]. Brachial artery access offers advantages for visceral artery revascularization and serves as an alternative in cases where the femoral approach is unsuitable due to prior endovascular or open procedures. Upper extremity access (UEA) via the brachial artery is routinely employed in several complex procedures, particularly those involving the incorporation of renal and splanchnic vessels, such as parallel grafts and branched and fenestrated endovascular procedures. Additionally, the instructions for using off-the-shelf branched stent grafts for treating thoracoabdominal aortic aneurysms recommend antegrade deployment of target vessel bridging stents via UEA. Furthermore, the treatment of aortoiliac occlusive disease, such as covered endovascular reconstruction of aortic bifurcation (CERAB), can be a potential route for addressing more severe occlusive disease extending to the renal artery level or completely occluded iliac lesions that cannot be accessed through the femoral artery [4,5]. Access through the upper extremity may lead to complications such as bleeding, hematomas, or thrombosis due to the smaller diameter of the brachial artery [6]. Both open surgical cutdown brachial access (OSA) and ultrasound-guided percutaneous brachial access (UPA) can achieve arterial access. However, percutaneous interventions have been associated with fewer complications in previous studies [7]. This study aims to compare the outcomes of OSA and UPA in treating peripheral vascular and aortic diseases.

## 2. Material and Methods

### 2.1. Study Design

This retrospective observational study investigates the outcomes of OSA and UPA in managing peripheral arterial and aortic disease. Data were collected from our tertiary academic referral center between 2019 and 2023. This study was approved by the institutional review board and the Ankara University Faculty of Medicine Human Research Ethics Committee (date: 28 August 2023, no. 2023/398).

Individuals over 18 years of age requiring brachial access for peripheral arterial or aortic interventions were eligible for inclusion. Patients with a previous vascular graft (e.g., patch) at the access site were excluded from this study. The physician reviewed the indications for brachial artery access and its viability at the time of the initial procedure, and no patients were excluded retrospectively based on the anatomy of the access vessel. Patients were excluded if the brachial access intervention was used for diagnostic purposes only.

Patient-specific variables were extracted from hospital records. Data included age, gender, body mass index (BMI in kg/m^2^), chronic obstructive pulmonary disease (COPD), smoking status, hypertension, cerebrovascular disease, diabetes mellitus (DM), coronary artery disease (CAD), and renal disease (estimated glomerular filtration rate < 60 mL/min/1.73 m^2^). Preoperative use of ACE inhibitors, beta-blockers, calcium antagonists, diuretics, acetylsalicylic acid, clopidogrel, and anticoagulants was also recorded. Additionally, procedural time, arterial access method, and sheath size were assessed.

### 2.2. Operative Technique

All procedures were performed by the same cardiovascular surgeons in a hybrid room with the aid of fluoroscopic guidance. The decision to use the brachial access route and the choice between the OSA and UPA methods were based on the physicians’ preferences. For percutaneous access, brachial artery punctures were performed at the olecranon fossa level to allow better vessel compression with ultrasound guidance [6]. After sheath placement, patients were heparinized to achieve a target activated clotting time (ACT) above 250 s. The patients regularly receiving oral anticoagulation therapy stopped treatment two days before the intervention, and the antiplatelet drugs were resumed upon hospitalization. No arterial closure devices were utilized.

The manual compression and immobilization of the elbow joint were maintained for 12 h in the UPA procedures. In the OSA method, the arterial puncture was closed with a 6-0 polypropylene suture in a separate manner. Clinical examinations were conducted at 1 h, 6 h, 1 day, 2 weeks, and 30 days after the procedure. During the follow-up period, decisions for re-exploration were made based on physician assessment in cases of hematoma, ischemia, and neuropathy. All the patients underwent duplex ultrasound examinations at 1 day, 15 days, and 30 days postoperatively. Dual antiplatelet therapy for at least 6 months or a combination of an oral anticoagulant and an antiplatelet regimen was prescribed postoperatively for both methods. 

### 2.3. Endpoints and Definitions

The primary endpoint was the complication rate at the access site 30 days postoperatively. Access-related complications included bleeding requiring re-exploration, acute upper limb ischemia, thrombosis, pseudoaneurysm, arteriovenous fistula, and nerve injury associated with brachial access.

The secondary outcomes included major adverse events and minor access-site vascular complications (infection, deep venous thrombosis, hematoma, arteriovenous fistula, and transient peripheral nerve injury) not requiring additional surgeries, as well as the type and incidence of open or endovascular adjunctive procedures at the access site. Major adverse events at 30 days included myocardial infarction, death, and cerebrovascular events. Stroke was defined as a new acute ischemic lesion diagnosed by cerebral imaging modalities. TIA was defined as a transient focal neurologic deficit with complete resolution within 24 h and no new ischemic lesion evident on cerebral imaging. Myocardial infarction was defined by ischemic symptoms such as chest pain, ST-segment elevation of >1 mm in two or more leads, and troponin levels more than twice the normal limit. 

### 2.4. Statistical Analysis

All statistical analyses were performed using SPSS version 28.0 (IBM). Continuous variables were expressed as mean ± standard deviation and categorical variables as counts and percentages. Continuous data were compared using Student’s *t* test, and categorical data were compared using Fisher’s exact test. An α-value of 0.05 was set as the threshold for statistical significance.

## 3. Results

In our single-center retrospective study, we analyzed 485 patients. The preoperative baseline clinical and demographic characteristics of the patients are presented in Table 1. Most of the patients were men (*n* = 182, 56.9% for UPA vs. *n* = 99, 60% for OSA), with a mean age of 67.2 years in UPA and 69.4 years in OSA. The mean BMI was similar in both groups. Preoperative data such as smoking, diabetes, COPD, and hypertension had similar rates when compared in both groups and were statistically insignificant (*p* > 0.05) (Table 1). We also compared the preoperative use of antiaggregants and anticoagulants in both patient groups, and the rates were similar and statistically insignificant (*p* > 0.05). Peripheral arterial diseases involving the common femoral artery, superficial femoral artery, and external iliac artery, as well as combinations of these locations, were the most common lesion treated (UPA *n* = 216 vs. OSA *n* = 45); the second most common intervention was for aortoiliac occlusive diseases (UPA *n* = 66 vs. OPA *n* = 36). Figure 1 represents the flowchart of the procedures performed on all patients.

Table 2 presents the angiographic and intraoperative data. There is a significant difference in operation times between the two groups. The mean operation time for the percutaneous procedure was 164.5 ± 45.4 min, while the mean operation time for the cutdown procedure was 289.2 ± 79.4 min. The difference is statistically significant (*p* = 0.003). The longer operation time for the cutdown procedure is due to the additional steps required at both the beginning and end of the procedure. The comparison of mean sheath size and contrast material used in both groups showed similar results and were statistically insignificant (*p* < 0.05). Table 2 also details the types of anesthesia and sheath sizes used in the procedures. At our institution, the preferred site for brachial access interventions, in both the percutaneous and cutdown groups, is typically the distal brachial artery, followed less frequently by the distal axillary and proximal brachail arteries (Table 2).The overall technical success rate was 95% without any differences between the UPA (94%) and OSA (97%) procedures (*p* = 0.41). 

Table 3 lists the complications, procedures performed for complications, and mortality in the patients. Of the 485 patients, 449 had complete follow-up data available 30 days postoperatively. A total of 3 patients died within 30 days after the procedure, with 2 deaths in the UPA group (0.6%) and 1 in the OSA group (0.6%) (*p* = 0.131). The cause of death was not related to the access site or the procedure. No myocardial infarction occurred in either group. A stroke occurred in two patients, one in the UPA group and one in the OSA group. One patient in the UPA group developed a TIA. 

Only two patients in the UPA group and one patient in the OSA group experienced infections at the intervention site. Nerve damage was observed in four patients in the UPA group and one patient in the OSA group (*p* = 0.045), with no permanent nerve injuries reported during follow-up. When comparing postprocedure hematoma rates, 15 patients in the UPA group and 2 in the OSA group developed hematomas, a statistically significant difference (*p* = 0.004). Thrombus formation occurred in nine patients in the UPA group and three in the OSA group, which was not statistically significant (*p* = 0.078). Pseudoaneurysm developed in 11 patients in the UPA group and 4 in the OSA group, with the difference not being statistically significant (*p* = 0.089). Overall, complications were observed in 32 (10%) of the UPA patients and 8 (4.8%) of the OSA patients, a statistically significant difference (*p* = 0.004). Reoperations were performed on 23 patients (7.2%) in the UPA group and 8 patients (4.8%) in the OSA group (*p* = 0.137). The procedures performed during the reoperations are detailed in Table 3. The mean length of hospital stay for the patients is also compared in Table 3. The 30-day mortality rate was 0.6% (2 patients) in the UPA group and 0.6% (1 patients) in the OSA group, with the difference not being statistically significant (*p* = 0.131).

## 4. Discussion

As endovascular interventions increase, surgeons continue to explore new sites for arterial access. In our study, patients who underwent interventions with brachial access were analyzed in two comparable groups: ultrasound-guided percutaneous (UPA) and open surgical incisional (OSA) approaches. UPA was associated with a higher incidence of access complications. We also reported that perioperative morbidity, including major adverse events and access site-related complications, was significantly lower in the OSA group. Additionally, there was a significant difference in operation times between the two groups. The mean operation time for the percutaneous procedure was 164.5 ± 45.4, while the mean operation time for the cutdown procedure was 289.2 ± 79.4. The difference is statistically significant (*p* = 0.003). This result is also emphasized by Bertoglio et al. This may be due to the complexity of endovascular procedures, the bigger sheath size, and the requirement of surgical exposure [4]. However, mortality rates were statistically insignificant between the UPA and OSA groups (*p* = 0.131). The brachial access approach may be controversial due to the potential risks of upper limb ischemia, nerve injury, and stroke. In our study, all complications rates were 32 (10%) vs. 8 (4.8%), *p* = 0.004, respectively. On the other hand, brachial access is crucial in cases where femoral access is contraindicated or technically difficult, such as after kissing stent and bifurcated graft placement [4,6]. Despite its minimally invasive nature, studies have reported elevated complication rates. In a study by Alvarez-Tostado et al., complications related to access sites were seen in 21 out of 323 patients (6.6%), with surgical intervention required in 13 patients. Cutdown was performed in 29 patients (9%), with no complications observed in those who underwent the cutdown procedure [7]. Madden et al. reported the use of percutaneous intervention primarily for diagnostic studies, presenting an access site complication rate of 10.6%, and concluded that an increased sheath size was associated with a higher risk of procedure complications [8]. Other studies have indicated that complications in brachial interventions increase with sheath size, but open surgical interventions decrease this complication rate [9]. Surgical incisions for brachial access may be appropriate before procedures requiring larger sheaths. To avoid complications, the smallest possible sheath should be used, and interventions should be performed with a cutdown incision if necessary. Stavroulakis et al. assessed the use of brachial access for endovascular treatment of iliac artery disease, demonstrating a 12% rate of access site-related adverse events [10]. Another group reported an 8.7% access-related complication rate in open brachial access for iliac stenosis treatment [11]. A literature analysis on OSA revealed lower access complication rates, ranging from 4% to 8%. Kret et al. also reported a 4.1% access complication rate after arterial cutdown, consistent with our study results of 35% for UPA vs. 13.3% for OSA [9]. Bertoglio et al. found some procedural variables such as left-side access (*p* < 0.001) were more frequently used in the percutaneous access group and elbow crease access (*p* < 0.001) in the open surgical group. However, no significant differences were found between the groups after propensity matching [4].

The primary endpoint, the access site complication rate, was significantly higher in the UPA group, despite the higher rate of 7 Fr sheath used in the OSA group. Reoperations due to occlusive complications were more frequently seen for surgical intervention. The level of sheath insertion may have a role in this AF [7,9]. The greater incidence of dual antiplatelet therapy in the UPA group could influence the results and increase the rate of access site complications. However, there were no statistically significant differences between the two groups regarding preoperative antiplatelet and anticoagulation therapy (for preoperative antiplatelet: 84% for UPA vs. 75.7% for OSA, *p* = 0.56; for anticoagulation: 14% for UPA vs. 12.1% for OSA, *p* = 0.609). Studies have not defined antiplatelet treatment as a potential risk factor for brachial access complications. Conversely, sheath size and female gender are known risk factors. Alvarez-Tostado’s study associated female gender with a higher complication rate (11.5% vs. 2.7% for males; *p* = 0.003) and higher brachial thrombosis rates [7]. Kret et al. demonstrated that male gender (*p* < 0.01), smaller sheath size (<5 Fr; *p* = 0.03), and arterial cutdown (*p* = 0.04) were protective factors against access site complications [9]. De Carlo et al. found a relationship between sheath size and access complication rate, with higher rates of brachial access complications in percutaneous interventions with larger sheaths [12]. Patients with polyvascular disease do not fall into a single category with uniform risk.

The femoral arterial route remains the preferred option for most peripheral arterial interventions. To reduce perioperative access-related complications, various vascular closure devices have been developed, though they are primarily indicated for femoral artery use and have been evaluated at the brachial artery site in only a few studies [13,14,15].

Currently, radial access is routinely used for the diagnosis and treatment of coronary artery disease [3,4,5]. However, this approach is not always suitable for lower-limb vascular lesions due to the distance from the access site to the target vasculature. Coscas et al. reported 24 procedures using radial access, with a 13% radial artery occlusion rate and 2 radial artery ruptures treated by manual compression [16]. The major limitation of radial access is the small diameter of the radial artery, which limits sheath sizes and poses a risk for artery occlusion [16]. Kiemenj et al. compared percutaneous coronary interventions using radial, brachial, and femoral access with a 6 Fr sheath, demonstrating significantly higher rates of major complications with brachial and femoral approaches compared to radial access (2.3%, 2.0%, and 0%, respectively; *p* = 0.035) [17]. New data are needed for patients who have suffered a stroke, aortic aneurysm, or peripheral artery disease (PAD). Additionally, there is a significant lack of data on subjects with polyvascular atherosclerotic disease, a population at very high risk. Accurate assessment is essential to stratifying risk in these patients. The possibility of effectively assessing cardiovascular risk in patients with polyvascular atherosclerotic disease using imaging techniques may be improved by correlating imaging data with metabolic targets or various medical therapies. This approach could verify the effectiveness of these therapies in preventing the progression of cardiovascular diseases (CVDs) [18]. Cerebrovascular complications remain a significant obstacle for upper extremity access. Although there is no difference in stroke rates between various UEA closure techniques, future studies should determine whether avoiding upper extremity access could significantly reduce adverse events during complex aortic endovascular procedures [19].

The major limitations of our study include its single-center, retrospective, and non-randomized design. It is not possible to report the number of patients for whom UEA was deemed infeasible during procedural planning by the physicians; thus, selection bias cannot be excluded. On the other hand, our center treats a huge number of patients annually and has substantial experience in obtaining upper extremity access. Furthermore, the option of a brachial arterial approach was at the discretion of the physicians, who routinely preferred the technique. Additionally, the study cohort includes patients who underwent a wide range of procedures using upper extremity brachial access, including arch and descending aorta repairs and treatments for infrarenal aortoiliac occlusive diseases. While this single-center retrospective study may have inherent biases, the physicians involved were experienced in both access techniques.

## 5. Conclusions

The results of this study have revealed that the incidence of brachial access complications is higher for UPA than for OSA in patients undergoing endovascular arterial interventions. Further research should include a large-scale, randomized controlled trial to validate the potential advantages of OSA.

## Figures and Tables

**Figure 1 jcm-13-04179-f001:**
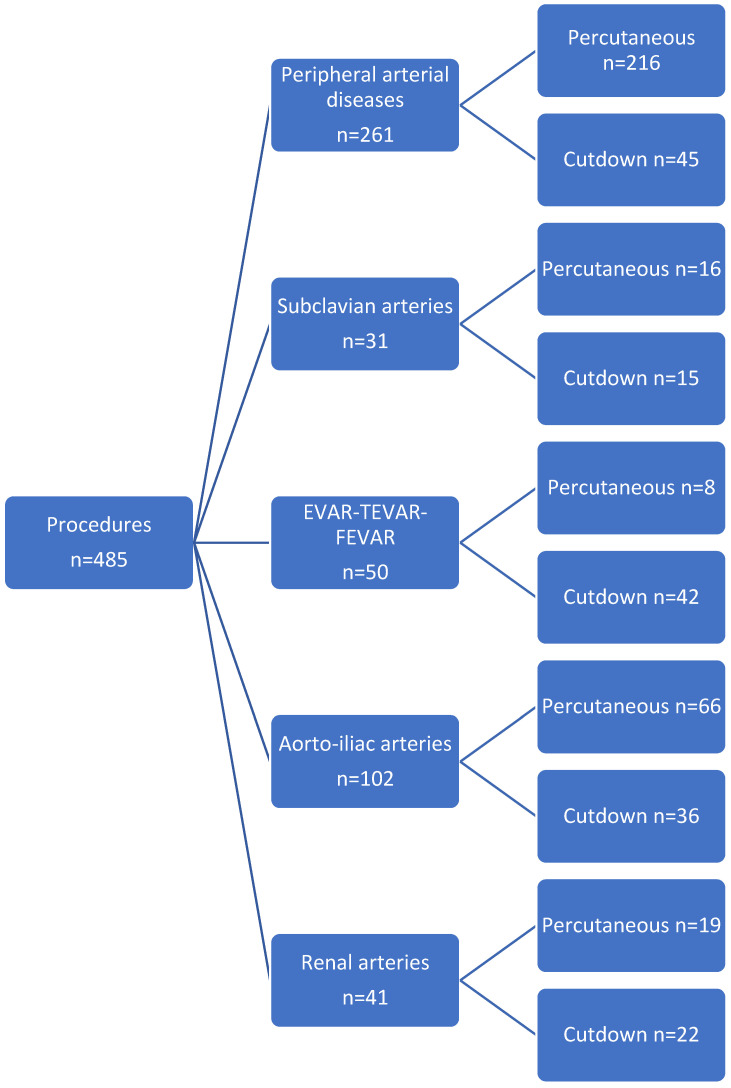
Flowchart of the procedures. EVAR = abdominal endovascular aneurysm repair; TEVAR = thoracic endovascular aneurysm repair; FEVAR = fenestrated abdominal endovascular aneurysm repair.

**Table 1 jcm-13-04179-t001:** Demographic data and comorbidities in patients.

	Percutaneous (*N* = 320)	Cutdown (*N* = 165)	*p*
Age (mean)	67.2	69.4	0.416
BMI (mean)	26.7	28.4	0.114
Male	182 (56.9)	99 (60.0)	0.246
Female	138 (43.1)	66 (40.0)	0.113
Smoking	149 (46.6)	104 (63.0)	0.300
Diabetes mellitus	183 (57.2)	92 (55.8)	0.149
COPD	172 (53.8)	103 (62.4)	0.172
Hypertension	208 (65.0)	118 (71.5)	0.345
Prior CABG or PCI	124 (38.8)	72 (43.6)	0.254
Dialysis	45 (14.1)	29 (17.6)	0.656
Preoperative antiaggregant	269 (84.1)	125 (75.8)	0.56
Preoperative anticoagulation	45 (14.1)	20 (12.1)	0.609

Data are presented as *n* (%) or mean ± standard deviation. BMI = body mass index; COPD = chronic obstructive pulmonary disease; CABG = coronary artery bypass graft surgery; PCI = percutaneous coronary intervention.

**Table 2 jcm-13-04179-t002:** Angiographic and intraoperative characteristics.

	Percutaneous *N* = 320	Cutdown *N* = 165	*p*
Procedure time (min)	164.5 ± 45.4	289.2 ± 79.4	0.003
Sheath size (mean) (French)	6.2 ± 1.7	8.1 ± 1.9	0.009
Contrast agent volume (mL)	55.6 ± 32.3	59.5 ± 36.7	0.008
Anesthesia			
Local	207 (64.7)	59 (35.8)	
Regional	0	13 (7.9)	
General	113 (35.3)	93 (56.4)	
Sheath diameter			
5 French	19 (5.9)		
6 French	219 (68.4)		
7 French	82 (25.6)	5 (3.0)	
8 French		134 (81.2)	
9 French		26 (15.8)	
Access or puncture site			
Distal axillary or proximal brachial	72 (22.5)	28 (17.0)	
Elbow crease or distal brachial	248 (77.5)	137 (83.0)	

Data are presented as *n* (%) or mean ± standard deviation.

**Table 3 jcm-13-04179-t003:** Complications and management.

	Percutaneous *N* = 320	Cutdown *N* = 165	*p*
Infection at access site	2 (0.6)	1 (0.6)	0.122
Nerve injury	4 (1.3)	1 (0.6)	0.045
Hematoma	15 (4.7)	2 (1.2)	0.004
Thrombosis	9 (2.8)	3 (1.8)	0.078
Pseudoaneurysm	11 (3.4)	4 (2.4)	0.089
Reoperation	23 (7.2)	8 (4.8)	0.137
All Complications	32 (10.0)	8 (4.8)	0.004
Management			
Endovascular	7 (2.2)	8 (4.8)	
Surgical			
Thrombectomy	24 (7.5)	4 (2.4)	
Hematoma drainage	21 (6.6)	8 (4.8)	
Patch repair	4 (1.3)	1 (0.6)	
Hospital stay (days)	8.43 ± 6.3	9.87 ± 8.78	0.006
Mortality (30-day)	2 (0.6)	1 (0.6)	0.131

Data are presented as *n* (%) or mean ± standard deviation.

## Data Availability

Data are contained within the article.

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
