# Peer review of "Adverse Events in Open Surgical vs. Ultrasound-Guided Percutaneous Brachial Access for Endovascular Interventions"

_jcm, 2024, doi:10.3390/jcm13144179_

Round 1

Reviewer 1 Report

Comments and Suggestions for Authors

Evren Ozcinaret al. present an original article with the aim to compare the incidence of access site complications between ultrasound guided percutaneous brachial access (UPA) and open surgical incisional brachial access (OSA) in management of peripheral arterial diseases. The article offers interesting data but key inputs need to be considered to improve the quality and scientific impact of the manuscript.

1. Abstract: Is the statement “… brachial artery access are commonly used for coronary procedures” verified? Radial and femoral artery are commonly used for coronary procedures, and not brachial artery.

2. It’s suggested to add more details on the types of cardiovascular procedures where we can use brachial access.

3. Please, add inclusion and exclusion criteria in the Material and Methods section. In addition, I suggest to add a specific figure on the study population.

4. I suggest to add in the Discussion the importance of the cardiovascular imaging in the risk assessment of the patients before the procedure and in view of the choice of the best brachial approach (UPA or OSA).
Please, add the following reference to improve the new sentences: Perone F, et al. Role of Cardiovascular Imaging in Risk Assessment: Recent Advances, Gaps in Evidence, and Future Directions. J Clin Med. 2023 Aug 26;12(17):5563. doi: 10.3390/jcm12175563.

5. Based on your results, what are the future studies and direction? Add this point in the discussion.

Author Response

Response to Reviewer 1 Comments

1. Summary

Thank you very much for taking the time to review this manuscript. Please find the detailed responses below and the corresponding revisions and corrections in track changes in the re-submitted files.

2. Questions for General Evaluation

Reviewer’s Evaluation

Response and Revisions

Does the introduction provide sufficient background and include all relevant references?

Can be improved

Thanks for the reviewer suggestions, introduction was revised.

Are all the cited references relevant to the research?

Can be improved

The references were also revised.

Is the research design appropriate?

Yes

Are the methods adequately described?

Can be improved

Method section was revised.

Are the results clearly presented?

Yes

Are the conclusions supported by the results?

Yes

3. Point-by-point response to Comments and Suggestions for Authors

Comments 1: Evren Ozcinar et al. present an original article with the aim to compare the incidence of access site complications between ultrasound guided percutaneous brachial access (UPA) and open surgical incisional brachial access (OSA) in management of peripheral arterial diseases. The article offers interesting data but key inputs need to be considered to improve the quality and scientific impact of the manuscript.

Abstract: Is the statement “… brachial artery access are commonly used for coronary procedures” verified? Radial and femoral artery are commonly used for coronary procedures, and not brachial artery.

Response 1:  Thank you for pointing this out. We agree with this comment. Therefore, we have revised the sentence for  misunderstandings. First of all, we aim to investigate access failure and adverse events associated with endovascular interventions performed via upper extremity access. We also aim to compare the open and ultrasound assisted percutaneous interventions in a specific brachial artery region. We revised the sentence :Abstract section-Background: line 12 ‘’While radial and brachial artery access are commonly used for coronary procedures, the common femoral artery remains the most frequent site for endovascular treatments due to better ergonomics and proven technical success.’’ And add the edited sentence in the abstract: ‘’Although radial artery access is commonly used for coronary procedures, the common femoral artery remains the most frequent site for endovascular treatments due to better ergonomics and proven technical success. Meanwhile, data on using upper extremity access via the brachial artery during complex endovascular aortic interventions are lacking.’’

And also, introduction section, the sentence in Line 36 was revised and brachial artery term was skipped fort he coronary interventions.

Comments 2:  It’s suggested to add more details on the types of cardiovascular procedures where we can use brachial access.

Response 2: Agree. We have, accordingly, revised the introduction section in Line 43-48 to emphasize the importance of upper extremity access via brachial artery. We revised the paragraph:’’Upper extremity access (UEA) is routinely employed in several complex procedures, particularly those involving the incorporation of renal and splanchnic vessels, such as parallel grafts and branched and fenestrated endovascular procedures. Additionally, the instructions for using off-the-shelf branched stent grafts for treating thoracoabdominal aortic aneurysms recommend the antegrade deployment of target vessel bridging stents via UEA. Furthermore, the treatment of aortoiliac occlusive disease, such as covered endovascular reconstruction of aortic bifurcation (CERAB), can be a potential route for addressing more severe occlusive disease extending to the renal artery level or completely occluded iliac lesions that cannot be accessed through the femoral artery.’’

Comments 3: Please, add inclusion and exclusion criteria in the Material and Methods section. In addition, I suggest to add a specific figure on the study population.

Response 3: Thank you for pointing out this.  In the Materials and methods section, 2.1 Study Design section briefly explain the inclusion and exclusion criteria in the line 62-64. And also added a paragraph:’’Patients with a previous vascular graft (e.g., patch) at the access site were excluded from the study. The physician reviewed the indications for brachial artery access and its viability at the time of the initial procedure, and no patients were excluded retrospectively based on the anatomy of the access vessel.’’ We planned to demonstrate study population in the flowchart.

Comments 4: I suggest to add in the Discussion the importance of the cardiovascular imaging in the risk assessment of the patients before the procedure and in view of the choice of the best brachial approach (UPA or OSA).
Please, add the following reference to improve the new sentences: Perone F, et al. Role of Cardiovascular Imaging in Risk Assessment: Recent Advances, Gaps in Evidence, and Future Directions. J Clin Med. 2023 Aug 26;12(17):5563. doi: 10.3390/jcm12175563.

Response 4: Agree. We have, accordingly, revised the Discussion section in Line 231. We emphasized the importance of cardiovascular imaging:’’ New data are needed for patients who have suffered a stroke, aortic aneurysm, or peripheral artery disease (PAD). Additionally, there is a significant lack of data on subjects with polyvascular atherosclerotic disease, a population at very high risk. Accurate assessment is essential to stratify risk in these patients. The possibility of effectively assessing cardiovascular risk in patients with polyvascular atherosclerotic disease using imaging techniques may be improved by correlating imaging data with metabolic targets or various medical therapies. This approach could verify the effectiveness of these therapies in preventing the progression of cardiovascular diseases (CVDs).’’

And also, we have added and cite the Reference:  ‘’Perone F, et al. Role of Cardiovascular Imaging in Risk Assessment: Recent Advances, Gaps in Evidence, and Future Directions. J Clin Med. 2023 Aug 26;12(17):5563. doi: 10.3390/jcm12175563’’as Ref. 18 for enhancing the paragraph.

Comments 5: Based on your results, what are the future studies and direction? Add this point in the discussion.

Response 5:  Agree. We have, accordingly, revised the Discussion section in Line 231 and go on. Cerebrovascular complications remain a significant obstacle for upper extremity access. Although there is no difference in stroke rates between various UEA closure techniques, future studies should determine whether avoiding upper extremity access could significantly reduce adverse events during complex aortic endovascular procedures.

4. Response to Comments on the Quality of English Language

Point 1: English language fine. No issues detected.

Response 1:    Thanks for the reviewer’s comment.

5. Additional clarifications

We have revised the manuscript according to the reviewer’s suggestions.

Reviewer 2 Report

Comments and Suggestions for Authors

This manuscript titled "Adverse Events in Open Surgical Versus Ultrasound Guided Percutaneous Brachial Access for Endovascular Interventions" has several unresolved issues and limitations that require significant revisions:

The manuscript lacks a clearly stated hypothesis and specific objectives. While it aims to compare complications between two techniques, it doesn't explicitly state what it seeks to prove or disprove beyond comparison.

If the study is retrospective, this should be clearly stated, as it impacts the level of evidence and potential biases.

The manuscript does not provide details on how patients were selected or matched for the two different procedures, raising concerns about potential selection bias.

There is no mention of controlling for confounding variables that could impact complication rates, such as patient comorbidities, operator experience, and procedural complexity.

The provided p-values (e.g., p = 0.003, p = 0.004) suggest significance, but without context or discussion on clinical relevance, these results might be misleading.

The mean operation times for the two procedures are significantly different (164.5 vs. 289.2 min), which could be due to the inherent differences between the procedures rather than a direct comparison of outcomes.

The manuscript lists raw numbers of complications but does not provide rates per 1000 procedures or similar metrics, which would offer a clearer comparison.

The reoperation rates are presented without context or analysis of why these rates might differ between groups.

The conclusion that ultrasound-guided percutaneous brachial access has higher complication rates may be an overgeneralization without considering other factors influencing outcomes.

The manuscript lacks a thorough discussion of the findings, including potential reasons for the observed differences, implications for clinical practice, and comparison with existing literature.

The background section is brief and does not adequately review relevant literature or provide a rationale for the study.

The manuscript does not include references to support the statements in the background or contextualize the findings within the broader scientific literature.

Addressing these issues would significantly improve the manuscript's rigor, validity, and overall quality.

Comments on the Quality of English Language

 Moderate editing of the English language is required.

Author Response

Response to Reviewer 2 Comments

1. Summary

Thank you very much for taking the time to review this manuscript. Please find the detailed responses below and the corresponding revisions and corrections highlighted changes in the re-submitted files.

2. Questions for General Evaluation

Reviewer’s Evaluation

Response and Revisions

Does the introduction provide sufficient background and include all relevant references?

Must be improved

Thanks for the reviewer’s recommendation. We have revised the introduction.

Are all the cited references relevant to the research?

Must be improved           

All the cited references were revised and added new references. 4,18,19

Is the research design appropriate?

Must be improved

Research design also expressed.

Are the methods adequately described?

Must be improved

Method section is also revised.

Are the results clearly presented?

Must be improved

Are the conclusions supported by the results?

Must be improved

Conclusion is also revised and edited.

 3. Point-by-point response to Comments and Suggestions for Authors

Comments 1: This manuscript titled "Adverse Events in Open Surgical Versus Ultrasound Guided Percutaneous Brachial Access for Endovascular Interventions" has several unresolved issues and limitations that require significant revisions:

The manuscript lacks a clearly stated hypothesis and specific objectives. While it aims to compare complications between two techniques, it doesn't explicitly state what it seeks to prove or disprove beyond comparison.

Response 1: Thank you for your valuable feedback. We have revised the manuscript to address your concerns regarding the hypothesis and specific objectives.The primary hypothesis of our study is that the ultrasound-guided percutaneous approach (UPA) for brachial access is associated with a higher rate of access-related complications compared to the open surgical approach (OSA). Our specific objectives are to compare the complication rates, procedural times, and outcomes between these two techniques in patients undergoing endovascular interventions for peripheral arterial and aortic diseases.

Comments 2: If the study is retrospective, this should be clearly stated, as it impacts the level of evidence and potential biases.

Response 2: Thanks for the reviewer’s suggestion. We have emphasized the retrospective study design material method section 2.1 segment in line 57. We agree the precious contribution. We clearly state the retrospective manner also in the study limitations. And also, Results section in line 114. We have emphasized the retrospective study design. We additionally revised the study limitations that may impact the level of evidence and potential biases.

Comments 3: The manuscript does not provide details on how patients were selected or matched for the two different procedures, raising concerns about potential selection bias.

Response 3: Thanks for the reviewer’s recommendations. We have revised the material method section in line 62-64: ’’ Individuals over 18 years of age requiring brachial access for peripheral arterial or aortic interventions were eligible for inclusion. Patients with a previous vascular graft (e.g., patch) at the access site were excluded from the study. The physician reviewed the indications for brachial artery access and its viability at the time of the initial procedure, and no patients were excluded retrospectively based on the anatomy of the access vessel. Patients were excluded if the brachial access intervention was used for diagnostic purposes only.’’.  And there is a potential for raising concerns about potential bias. However, study limitations principally resides its retrospective nature. We also revised and edited the study limitations section for a possible selection bias. On the other hand, the physicians involved were experienced with both access techniques.

 Comments 4: There is no mention of controlling for confounding variables that could impact complication rates, such as patient comorbidities, operator experience, and procedural complexity.

Response 4: Thanks for the reviewer’s suggestions. All procedures were performed by the same cardiovascular surgeons in a hybrid room. As the same team did all the interventions, there is no difference between operator experience. Procedural complexity is an issue. As we figured out in flowchart, 485 patients were involved. The procedural complexity was expressed in the Result section line 121-126. The comorbidities of the patients were illustrated at Table1, but there is no significant differences.

Comments 5:  The provided p-values (e.g., p = 0.003, p = 0.004) suggest significance, but without context or discussion on clinical relevance, these results might be misleading.

Response 5: Agree. Discussion section was revised and edited. As the reviewer mentioned, provided p values were expressed in the discussion. Discussion section line 179-183.

Comments 6: The mean operation times for the two procedures are significantly different (164.5 vs. 289.2 min), which could be due to the inherent differences between the procedures rather than a direct comparison of outcomes.

Response 6: Agree. We mentioned this issue in the Discussion section. We expressed the mean operation times for the two procedures. This is involving the open surgical access period as it is a time consuming intervention comparing with percutaneous procedures. And also, sheath size enlargement may affect the operation time of the procedure.

Comments 7: The manuscript lists raw numbers of complications but does not provide rates per 1000 procedures or similar metrics, which would offer a clearer comparison.

Response 7: Agree. Thanks for the reviewer’s recommendations. We have revised the tables which would offer a clearer comparison.

Comments 8: The reoperation rates are presented without context or analysis of why these rates might differ between groups.

Response 8: Agree. In the result section , reoperation rates were revised and emphasized. 

Comments 9: The conclusion that ultrasound-guided percutaneous brachial access has higher complication rates may be an overgeneralization without considering other factors influencing outcomes.

Response 9: Agree. As the multicenter international registries for upper extremity access genrally emphasize the percutaneous UEA is burdened by increased access failure versus surgical exposure. There are some concerns about routine percutaneous UEA and require further investigation. Bertoglio L et al. presented and we have added the article into the references.

Comments 10: The manuscript lacks a thorough discussion of the findings, including potential reasons for the observed differences, implications for clinical practice, and comparison with existing literature.

Response 10: Thanks for the reviewer’s recommendations. We have added references 4,18 and 19. And also, by the help of the comments, adding potential reasons for observed differences and implications for practice and comparison with existing literature.

Comments 11: The background section is brief and does not adequately review relevant literature or provide a rationale for the study.

Response 11: Thanks for the suggestion. The background section was revised and edited according the reviewer’s recommendations.

Comments 12: The manuscript does not include references to support the statements in the background or contextualize the findings within the broader scientific literature.

Response 12: According to the suggestions, references were revised and edited.

Comments 13: Addressing these issues would significantly improve the manuscript's rigor, validity, and overall quality.

Response 13: Thanks for the reviewer’s suggestion. The manuscript was revised and edited. Background, discussion sections were also reexamined for improving the overall quality.

4. Response to Comments on the Quality of English Language

Point 1: Moderate editing of English language required.

Response 1: Thanks for the reviewer’s suggestion. The text was revised and edited.

5. Additional clarifications

According to the suggestions of the reviewer, the manuscript was improved for overall quality.

Round 2

Reviewer 1 Report

Comments and Suggestions for Authors

The authors responded satisfactorily to my requests and comments, congratulations

Reviewer 2 Report

Comments and Suggestions for Authors

I recommend accepting this manuscript as my comments and suggestions are addressed adequately. Kindly check throughout the text, especially tables, to replace commas with points or decimals.